

# Widely assumed phenotypic associations in *Cannabis sativa* lack a shared genetic basis

Daniela Vergara[1], Cellene Feathers[1], Ezra L. Huscher[1], Ben Holmes[2], Jacob A. Haas[3] and Nolan C. Kane[1]

[1] Ebio, University of Colorado at Boulder, Boulder, CO, USA
[2] Centennial Seeds, Lafayette, CO, USA
[3] DabLogic, Denver, CO, USA

Corresponding author
Daniela Vergara,
daniela.vergara@colorado.edu

## ABSTRACT

The flowering plant *Cannabis sativa*, cultivated for centuries for multiple purposes, displays extensive variation in phenotypic traits in addition to its wide array of secondary metabolite production. Notably, *Cannabis* produces two well-known secondary-metabolite cannabinoids: cannabidiolic acid (CBDA) and delta-9-tetrahydrocannabinolic acid (THCA), which are the main products sought by consumers in the medical and recreational market. *Cannabis* has several suggested subspecies which have been shown to differ in chemistry, branching patterns, leaf morphology and other traits. In this study we obtained measurements related to phytochemistry, reproductive traits, growth architecture, and leaf morphology from 297 hybrid individuals from a cross between two diverse lineages. We explored correlations among these characteristics to inform our understanding of which traits may be causally associated. Many of the traits widely assumed to be strongly correlated did not show any relationship in this hybrid population. The current taxonomy and legal regulation within *Cannabis* is based on phenotypic and chemical characteristics. However, we find these traits are not associated when lineages are inter-crossed, which is a common breeding practice and forms the basis of most modern marijuana and hemp germplasms. Our results suggest naming conventions based on leaf morphology do not correspond to the chemical properties in plants with hybrid ancestry. Therefore, a new system for identifying variation within *Cannabis* is warranted that will provide reliable identifiers of the properties important for recreational and, especially, medical use.

## INTRODUCTION

Phenotypic variation within and between populations is an important characteristic to consider for classification purposes. In particular, if phenotypic variation exists between different lineages within a species, then characterizing trait correlations can shed light on how they are inherited, whether they are controlled by the same genes, and if they can be used for taxonomic purposes. Wild populations may carry shared ancestral traits despite being independently inherited, and therefore it may appear as if these traits were associated. Additionally, selection may favor certain trait combinations.

The angiosperm species *Cannabis sativa* has been cultivated for millennia for a range of purposes (*Li, 1973*, *1974*; *Russo, 2007*) and is currently, by some estimates, one of the world's most valuable crops (*Hutchison et al., 2019*). Unfortunately, widespread legal issues have hindered *Cannabis* research.

One of the most notable characteristics of the *Cannabis* plant is its chemistry: the production of a family of molecules known as cannabinoids which are mainly produced and stored in the trichomes of female flowers (*Gagne et al., 2012*; *Sirikantaramas et al., 2005*). The most studied of these cannabinoids are cannabidiolic acid (CBDA), and delta-9-tetrahydrocannabinolic acid (THCA), which are produced by the enzymes CBDA and THCA synthases, respectively. These two synthases are found at the final stage of the biochemical pathway, along with cannabichromenic acid synthase, a third less well-studied synthase, that produces Cannabichomenic acid (CBCA; *Page & Stout, 2017*). These three synthases use the same precursor molecule, cannabigerolic acid (CBGA; *Laverty et al., 2019*; *Page & Boubakir, 2014*; *Vergara et al., 2019*). The genetic sequences for the three synthases are very similar and at least the genes encoding CBDA and THCA synthase are close in proximity (*Weiblen et al., 2015*), suggesting they may have originated from the same ancestor gene (*Onofri, De Meijer & Mandolino, 2015*; *Padgitt-Cobb et al., 2019*). Additionally, in vitro, each of these synthases can produce at least eight different compounds including THCA and CBDA in different ratios (*Kovalchuk et al., 2020*; *Zirpel, Kayser & Stehle, 2018*). These enzymes may be classified as "promiscuous enzymes" due to their considerable similarities, the fact they act on the same precursor molecule, and they can produce each other's compounds (*Auldridge, McCarty & Klee, 2006*; *Chakraborty et al., 2013*; *Franco, 2011*).

When heated, THCA and CBDA are converted into the neutral forms THC and CBD (*Russo, 2011*), which interact with the human endocannabinoid system (*Pertwee, 1988*, *1997*, *2004*). Both THC and CBD have medicinal (*Russo, 2011*; *Swift et al., 2013*; *Volkow et al., 2014*) and economic value (*Evans, 2013*; *Kirsch, 2018*), but THC has been intensely selected by breeders and growers (*Volkow et al., 2014*) due to its psychoactive effects (*ElSohly & Slade, 2005*). Studies have also found CBDA may have medicinal benefits in its acidic form (*Takeda et al., 2008*, *2012*). Differences in leaf size, leaf shape, plant size, and inflorescence size are used in the *Cannabis* industry to categorize plants and these morphological differences are thought to be useful predictors of cannabinoid content.

Currently recognized lineages within the genus *Cannabis* include the narrowleaf drug types, *C. sativa* ssp. *sativa*, the broadleaf drug type *C. sativa* ssp. *indica*, the northern Eurasian wild *C. sativa* ssp. *ruderalis*, and at least one lineage of hemp (*Clarke & Merlin, 2013*). Among these subspecies, there is substantial phenotypic variation in the production of multiple cannabinoids (*McPartland & Russo, 2001*; *Russo et al., 2008*; *Russo & McPartland, 2003*) and terpenoids (*De la Fuente et al., 2020*; *Orser et al., 2017*; *Reimann-Philipp et al., 2019*), substantial genotypic variation (*Kovalchuk et al., 2020*; *Lynch et al., 2016*; *Sawler et al., 2015*; *Vergara et al., 2016*), and observed morphological variation in traits such as branching, internode length, and flowering time (*Clarke & Merlin, 2013*).

The main classifications used in the modern *Cannabis* industry parlance are "indica", "sativa", and "hybrids". Sativa plants are described as tall with narrow leaves and

lighter density buds, allegedly producing high levels of THCA, and therefore have uplifting and stimulating psychedelic effects after consumption. Indica plants are described as short with broad leaves and dense buds, and produce high levels of both THCA and CBDA believed to produce a relaxing effect (*Clarke & Merlin, 2013*; *McPartland, 2017*; *Vergara et al., 2016*). Yet, the associations between these multiple traits have not previously been researched. Other popular ideas suggest the important distinctions between *Cannabis* lineages related to the effects after consumption are due to differences in terpene profiles rather than to cannabinoid profiles. It may be that terpene profiles are more relevant to differences in *Cannabis* lineages than cannabinoids (*De la Fuente et al., 2020*; *Orser et al., 2017*; *Reimann-Philipp et al., 2019*). Crosses between "sativa" and "indica" plants are referred to as "hybrids" and these have variable phenotypes usually intermediate to the parents (*Vergara et al., 2016*). Finally, the hemp group has been traditionally used for industrial purposes such as fiber or oil production, however the legal definition of hemp includes any *Cannabis* plant with less than 0.3% THC by weight.

The colloquial naming convention of "indica" and "sativa" do not correspond to the scientific subspecies with similar names. Furthermore, these common distinctions do not reflect evolutionary relationships (*Sawler et al., 2015*; *Schwabe & McGlaughlin, 2019*; *Vergara et al., 2016*). This misidentification can be particularly problematic for medical patients who are depending on reliable and consistent products.

*Cannabis* is dioecious (*Divashuk et al., 2014*; *Van Bakel et al., 2011*), although monoecious plants exist, particularly in the hemp lineage (*Hillig, 2005*; *Peil et al., 2003*). Dioecious varieties are common for medicinal and recreational purposes, and selection in domestication has been focused on females due to the production of cannabinoids (*Gagne et al., 2012*; *Sirikantaramas et al., 2005*), with strong selection against males and hermaphrodites. Many commercially important traits are expressed at maturity, and if breeders could predict their late-stage expression through correlations among these traits earlier during development, selection could be made sooner, accelerating breeding cycles. Also, if traits early in the development allowed for distinguishing between sexes, males could be culled before pollen production and potential female pollination. This is important because females would undesirably divert energy to seeds instead of cannabinoids after being pollinated (*Clarke & Merlin, 2013*).

In this study, we quantified 18 phenotypic traits of 297 individuals from a first-generation backcross (BC1) between a female "Carmagnola" hemp and a male marijuana-type plant "Afghan Kush". Many of the morphological traits we measured are considered important by the *Cannabis* industry to characterize different plants. We predicted these traits would vary in the BC1, and therefore allow us to investigate the pattern of possible genetic correlations. Furthermore, we determined whether the association between morphological traits and cannabinoid chemistry could be used to characterize *Cannabis* lineages. Given that the current nomenclature is not supported scientific research (*Lynch et al., 2016*; *Sawler et al., 2015*; *Vergara et al., 2016*), it is possible that erroneous associations between morphological traits has contributed to the misnaming issues in the *Cannabis* industry.

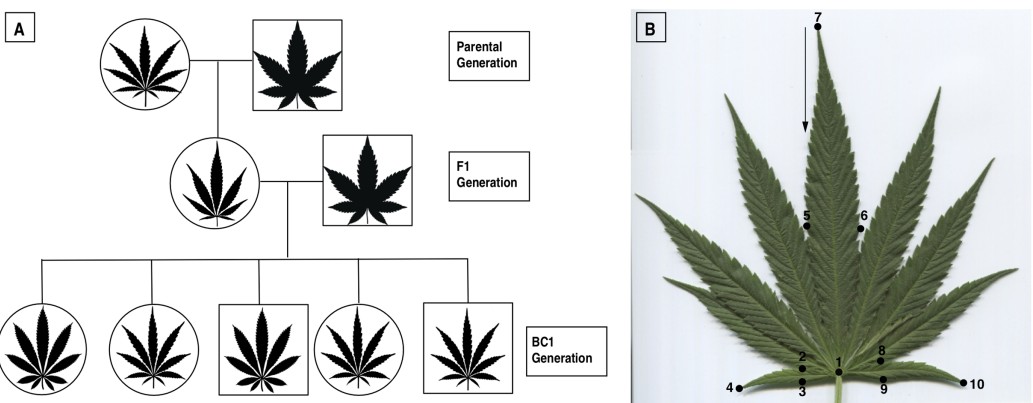

**Figure 1 Pedigree and landmarks.** (A) Pedigree of a first-generation backcross (BC1) between a male marijuana-type Afghan Kush and a female Carmagnola hemp. The F1 generation was backcrossed with a brother from the original male Afghan Kush to produce the BC1 Generation. (B) Exemplar leaf depicting the 10 points used for leaf shape analysis. The 10 points measured the first, central, and last leaflets.

## METHODS

### BC cross and measurements

A cross was performed between a female, narrowleaf "Carmagnola" hemp plant and a male broadleaf "Afghan Kush" plant at Centennial Seeds in Lafayette Colorado (Fig. 1A). One of the female first filial (F1) offspring of this cross was backcrossed to a male sibling of the parental male (Fig. 1A). Two hundred ninety-seven individuals from this backcross population (BC1; Fig. 1A) were started indoors on April 23, 2015 and on June 6, 2015, when the plants were 6 weeks old, they were transplanted outdoors in a field in Boulder County, Colorado.

Morphological measurements including height, stalk diameter, inner-node length, petiole length, leaf length and width, among other measurements, were obtained at two different time points during the growing cycle (Table S1). We chose these two time points, one at the beginning and one at the end of the growing season, to provide information on possible trait associations during the plant's development. The initial timepoint (IT) was taken at 6 weeks old (June 6, 2015), and the final timepoint (FT) at 19 weeks old (September 2, 2015) which corresponds to the beginning of the fall season. Additional traits were measured at the FT including bud count, size of biggest bud, length of longest branch, and number of buds on the longest branch (Table S1). At the FT, we also scanned a representative leaf from each plant. Therefore, we had fully extended longest leaf (FELL) measurements from both the IT and FT. At the FT, we determined the sex of all 297 individuals, and also measured the concentrations of three cannabinoids—THC, CBD, and CBG—from 100 plants. Here we removed plants identified as male to avoid pollination, after the representative leaf was scanned. Therefore, some of the measurements were taken on fewer individuals than at the IT.

### Phenotypic trait statistical analyses

To understand the change in individual phenotypes through the growing season, we calculated the difference between the initial measurements and the final measurements

(delta Δ) for some traits. Specifically, we calculated Δ for the four traits that were measured at both timepoints (Table S1). We then use these data to estimate the within-time point correlations for both the IT and the FT, and between-time correlations. All correlations were corrected with Bonferroni for multiple comparisons (*Weisstein, 2004*). Finally, we established whether any of the measured traits differed between males and females using *t*-tests with sex as the explanatory variable.

## Leaf shape analysis

We carried out a geometric morphometric analysis to develop a quantifiable measure of leaf shape. Specifically, we placed landmark coordinates on each leaf picture with the program TPS Dig2 (*Rohlf, 2006*). We used ten landmarks from the first, central, and last leaflet (Fig. 1B) which covers the whole leaf structure. Additionally, we measured the length and width of each of the leaves, counted the serration number in the center leaflet, and counted the number of leaflets.

We used the R package *Geomorph* (*Adams & Otárola-Castillo, 2013*) for all geometric morphometric analyses, following the methodology of *Vergara et al. (2017b)*. A Procrustes analysis was used to remove variability caused by position, orientation, and size and to quantify shape variation by superimposing the objects in a joint coordinate system. Then, a Principal Component Analysis (PCA) was used to identify the orthogonal structure in the data and to visually explore morphological variation among individuals.

We performed multiple statistical tests to understand whether leaf shape was related to any of the other measured traits at both timepoints and between timepoints. First, we implemented several multivariate analyses of variance (MANOVAs) with shape as the response variable for each of the measured traits in both timepoints and Δ. We then performed MANOVA models within each timepoint and Δ to understand whether the main effects of each trait affected leaf shape. We corroborated the results using multivariate multiple regressions.

## Cannabinoid concentration measurements

The concentrations for the three cannabinoids—CBG, THC, and CBD—were measured using gas chromatography on an SRI 86106 equipment with an MXT-35 column using 197–209 mg of dried flower as described in *Brenneisen & ElSohly (1988)*. When heated, the acidic compounds CBGA, THCA, and CBDA are turned into the neutral forms CBG, THC, and CBD, which is the reason why gas chromatography only quantifies the neutral forms of the compounds.

Given that the production of these three cannabinoids may be correlated because they are part of the same biochemical pathway (*Page & Boubakir, 2014*; *Page & Stout, 2017*; *Vergara et al., 2019*) and both CBDA and THCA synthases compete for the same precursor molecule –CBGA–, we analyzed them using a PCA to account for multicollinearity and to avoid redundancies. We used a K-means cluster analysis on PC1 vs PC2 to visualize the different cannabinoid groups. We also added the total cannabinoid concentration and measured the ratio of each cannabinoid over this total concentration (Table S1).
## Statistical Analyses

We examined the associations between the production of each cannabinoid and each of the measured traits at both timepoints and the Δ. We used cannabinoids as the explanatory variables for several MANOVA models to determine whether cannabinoid production explained differences among the measured traits. We corroborated the MANOVA results with multivariate multiple regressions, and correlated leaf shape to cannabinoid content to understand whether any association exists between those traits. Finally, we generated a variance-covariance matrix to establish the association within and between all phenotypic traits.

These data were added to the dryad repository (https://doi.org/10.5061/dryad.6t1g1jwxh). Statistical analyses, including leaf geometric morphometrics, were done using R (*R Core Team, 2013*) and the associated code is available on github (https://bit.ly/38DpE8D). All figures were generated in the R Studio platform version 1.1.383 (*R Core Team, 2013*) and enhanced with Adobe Illustrator 2019 (v23.0.6).

# RESULTS

## Phenotypic trait statistical analyses (including males and females)

Our results show that some phenotypic traits from the IT (Table S2) are correlated with each other after correcting with Bonferroni for multiple comparisons. For example, height is significantly correlated to the number of branches and the number of nodes even though these two traits are not significantly correlated to each other (Table S2). The positive correlation between traits related to height such as number of nodes and number of branches is expected. In other words, it is expected that tall plants will have multiple branches and nodes. It is also expected that traits that are not related to height, such as leaf-related characteristics, lack a significant correlation.

Similarly, the FT also shows that some traits are correlated at this stage (Table S3). Some of the height-related traits show a significant correlation. For example, tall plants have long side branches as well as thicker stalks. However, as expected, some traits lack association, such as stalk diameter and inflorescence number or size.

However, many of the significant associations within either the IT or FT are lost when both timepoints are correlated between them (Table S4). These various phenotypic traits are not predictive between time periods (Table S4); whether a young plant is tall or short is not indicative of the adult plant's height, and thus plants that are tall at the IT are not always the same ones that are tall at the FT. Therefore, we cannot establish whether, for example, tall plants also have thick stalks and numerous nodes since traits which are correlated while young, are not significantly correlated while adults. In other words, the plants exhibited different patterns of growth, irrespective of their initial size at the beginning of the growing season. The lack of correlations between the timepoints suggests that some of the statistically significant correlations may not be due to true biological variation but instead due to chance and to the multiple comparisons, despite correcting with Bonferroni.

The lack of significance between the ∆ correlations when compared to either the IT or FT (Tables S2 and S3) suggest that some of these correlations may be spurious. The non-significant correlations between the traits and their ∆ indicates that the measured characteristics do not follow a trend as they change over time. Therefore, the changes during the plant's lifetime impede future phenotypic predictions and the initial plant measurements cannot be used as an indication of future success as an adult, or how the phenotype will change during the plant's lifetime.

Similarly, these phenotypic traits are not different between males and females (Table S5). In other words, males cannot be distinguished from females with any of the physical characteristics that we measured in this study. However, some trait correlations do differ between the sexes (Table S6), but again are not consistent between the timepoints. The only significantly different trait between both groups is the number of buds in the main branch, which was taken at the FT, where males have a larger average number of buds (35.75) compared to females (24.80). However, this comparison is between only four males and 19 females, as most most males were removed from the field before these measurements were taken, and therefore this result may again not hold any true biological meaning.

## Leaf shape analysis

Our geometric morphometric analysis on leaf shape revealed that 82.3% of all variation in leaf shape is explained by the two first principal components (Fig. 2). The deformation grids in the top left and bottom right corners show the extreme trends in leaf morphologies. Even though these morphologies are not seen in any individual plant, these are the tendencies of the leaves in these furthest points of the morphospace. The individuals on the top-left side of the morphospace tend to have shorter and broader leaves, and as seen in the deformation grid, the first and last leaflets are pointing outwards. On the other hand, individuals in the bottom right side of the morphospace tend to have a longer middle leaflet, and the first and last leaflets are clumped together pointing downward. Our morphometric analysis shows no significant relationship between leaf shape and the plant's sex. Therefore, both male and female plants can have similar leaf shapes.

Our results suggest there are some trait correlations that describe leaf shape, but these are not correlated to growth rates, plant size, branching architecture, phytochemistry, or plant sex (Fig. S1). It appears that there could be a within-leaf effect because the FELL measurements correlate within them in the IT, and serration, leaf length, and number of leaflets correlate with leaf shape in the FT. However, the leaf measurements show no association between timepoints (Table S7).

The overall trend shows leaf shape is not explained by any of the plant traits measured on either timepoint (Table S7). The lack of association between a particular leaf shape and plant height, or any of the other plant traits, suggests that a tall plant can have broad or narrow leaves or high or low cannabinoid levels.

Furthermore, the MANOVA models with leaf shape as the response variable (PC1 and PC2, Table S7) show that the only traits that may explain leaf shape correspond to other leaf measurements which are leaf length, serration, and the number of leaflets from

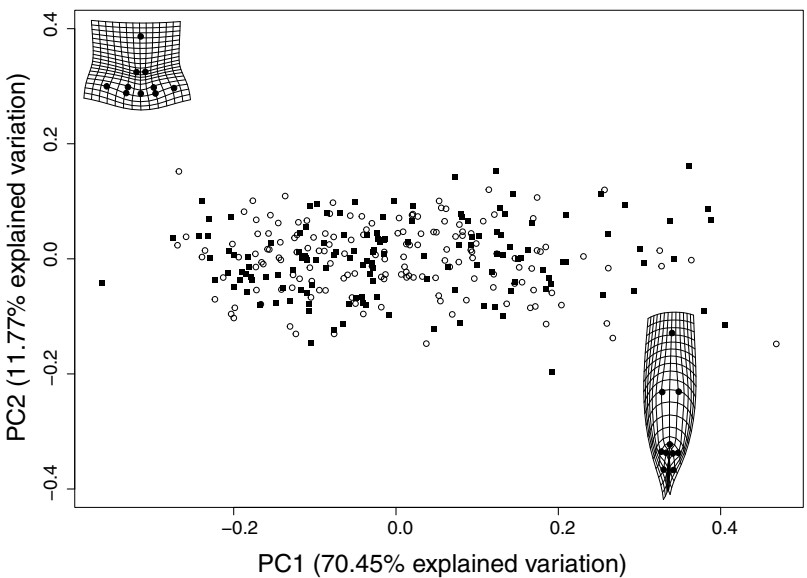

**Figure 2 Geometric morphometric analysis of leaf shape.** The two first PC explain 82.3% of the leaf shape variation, which is not related to sex (males are squares, females are open circles). The deformation grids mostly show the leaf deformations on PC1 which contains most of the variation. The grids show that individuals in the top left have squatter, broader leaves with the first and last leaflets pointing outwards, while those on the bottom right side of the morphospace have thinner leaves with a long mid-leaflet and the two outer leaflets lumped together pointing downwards.

that same leaf at the FT (Fig. S1). These MANOVA results confirm the within-leaf effect where these measured leaf traits are associated within the same leaf. The additional models that include the main effects of multiple traits support the individual MANOVA results given that no significant trend in any other phenotype explains leaf shape in the IT. The FT results of the within-leaf effect are confirmed with a further model including all of the leaf traits (leaf length: Wilk's $\lambda = 0.975$, $F = 3.560$, $P = 0.029740$; serration: Wilk's $\lambda = 0.967$, $F = 4.625$, $P < 0.012$; no. leaflets Wilk's $\lambda = 0.8105$, $F = 32.483$, $P = 2.1\text{e}{-}13$). However, the only significant interaction effect was between leaf length and leaf width (Wilk's $\lambda = 0.967$, $F = 4.739$, $P < 0.00095$). The multivariate multiple regressions confirmed the MANOVA results.

## Cannabinoid concentration measurements

Two clear chemotype clusters were identified in the PCA of cannabinoid chemistry determined by the antagonistic relationships between THC vs CBD and CBG (Fig. 3). PC1 and PC2 explain 94.2% of the variation and the two distinct groups identified in Fig. 3 (represented in triangles and diamonds) covary in a pronounced way, which show covariation of the original variables with two clear chemotype clusters (Fig. S2).

The positive loadings of both CBD and CBG on PC1 (0.71 and 0.70, respectively) compared to the negative loading of THC (−0.02) indicates that when CBD and CBG tend to increase, THC decreases and covaries in a different direction. Therefore, the value of PC1 increases when CBG or CBD increase. However, on PC2, both CBG and THC have a

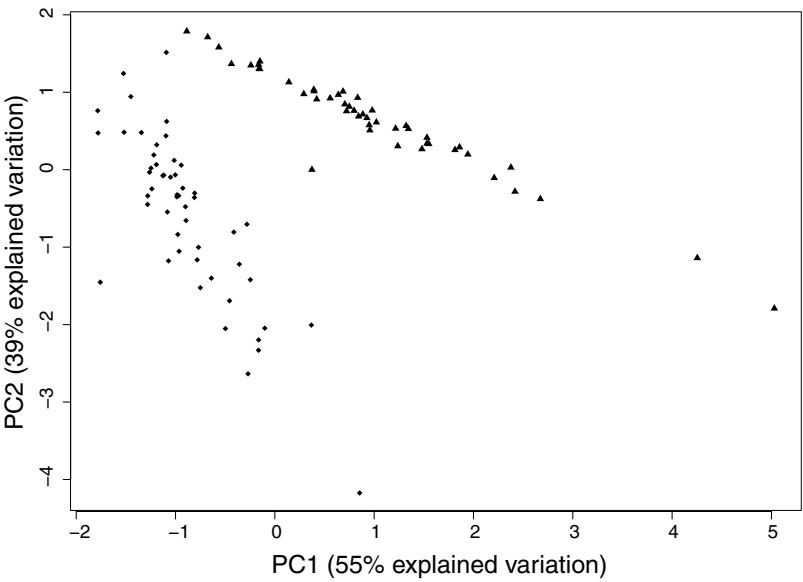

**Figure 3 PC1 and PC2 for cannabinoid variation.** PC1 and PC2 explain 94.2% of the overall cannabinoid variation for the three cannabinoids measured. There are two clear groups in the graph, squares and triangles, which correspond to the two clear cannabinoid clusters. The overall trend shows that the squares have low CBG and CBD and high THC. The triangles show high CBD and low THC.

negative loading (−0.31 and −0.91 respectively), indicating a high association, while CBD has a positive loading (0.28). PC2 is primarily determined by THC given its high loading value. In both PC1 and PC2, CBD and THC go in different directions.

Cannabinoid content showed no correlation with any of the other measured phenotypic traits at either time point (IT or FT) nor with the Δs (Table S8). These results were confirmed with the MANOVAs and multivariate multiple regressions.

## Leaf shape vs cannabinoid content

We found no relationship between leaf shape and cannabinoid content using PC1 for leaf shape and PC1 for cannabinoid variation (Fig. 4). Therefore, leaf shape is not predictive of cannabinoid content, and individuals that are high in a particular cannabinoid can have elongated or short leaves.

## DISCUSSION

In this study we examined correlations among various phenotypic traits from a morphologically diverse first-generation backcross (BC1) population to understand whether these multiple traits covaried with each other. We interpret these patterns of correlations as being genetically-based due to our development of a diverse array of progeny of known parentage all grown in a common environment. Our results suggest these traits are not constrained by strong genetic correlations and the initial associations between the various morphological traits in the parent generation can be broken by recombination. The lack of apparent genetic correlation between these traits suggest they can be selected for independently. Therefore, these traits are inherited independently,

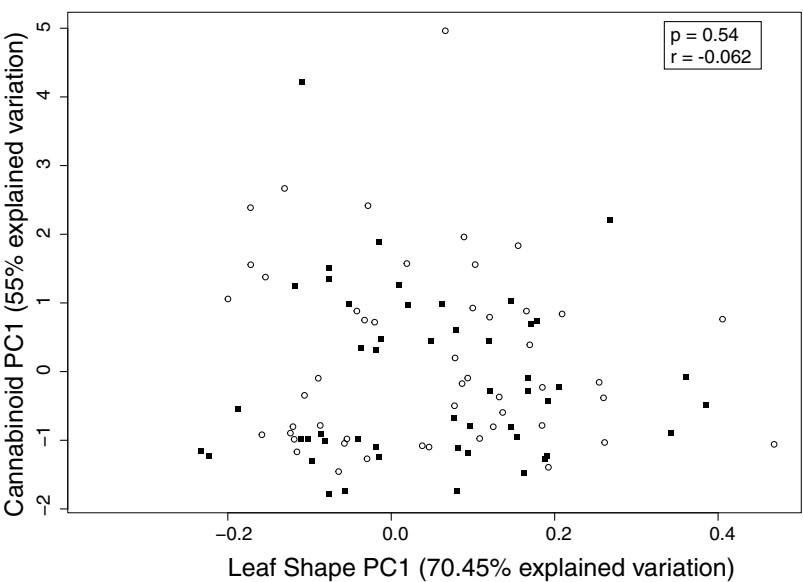

**Figure 4 Correlation between leaf shape (PC1) and cannabinoid variation (PC1).** Leaf shape is not significantly to cannabinoid variation even at the most extreme points in the morphospace. Males are shown in squares and females in open circles.

and would evolve separately unless selection acts to increase or maintain correlations among them. The lack of genetic correlations between the morphological traits was also reflected in a variable pattern of growth across the growing season. The dramatic trait changes and the distribution variation over the course of plant growth and development (*Coleman, McConnaughay & Ackerly, 1994*) may explain the lack of correlations between the two timepoints. This lack of correlation could also signify phenotypic plasticity which is common in plants and may be a form of adaptation (*Schlichting, 1986*; *Sultan, 1995*). Finally, in other species such as sugarcane, yield has not been associated to characteristics in the stalk (i.e., length, weight, diameter, number) nor to other traits such as plant height (*Aitken et al., 2008*; *Kang, Miller & Tai, 1983*), so these patterns are likely not unique to *Cannabis*.

The lack of sexual dimorphism in the measured traits for this study may be specific to this population, and particularly the measurements in the FT may be problematic due to the lack of males. Theoretical models suggest differences between males and females particularly in wind-pollinated plants (*Friedman & Barrett, 2009*). Additionally, sex differences in *Cannabis* have been found in traits not measured here, such as photosynthetic rates and senescence (*Dzhaparidze, 1969*; *Geber, Dawson & Delph, 2012*). Future studies may include these traits to examine differences between the sexes.

The PCA analysis facilitates the examination of shape variation for each structure independently (*Adams, Rohlf & Slice, 2004*), allowing us to distinguish differences in leaf shape (Figs. 1B and 2). As size is removed during the Procrustes superimposition, it does not determine the variation of the first principal component (PC1) as it does in traditional morphometrics, assuring that the main source of variation explored is shape. With this geometric morphometric analysis, we found that leaf shape is not related to sex
(Fig. 2), cannabinoid production (Fig. 4), or to multiple other phenotypic traits (Table S8), suggesting all of these traits segregate independently. However, we did find within-leaf associations between shape, leaf length, serration, and number of leaflets (Fig. S1) suggesting that within a single leaf some characteristics may be related to each other.

Because THC and CBD have attracted the most research and popular attention of all the cannabinoids and their synthases both compete for the same precursor CBGA, the relationship between these compounds revealed significant patterns. PC1 and PC2 for cannabinoid variation (Fig. 3) explain 94.2% of the variation due to the fact that there are only three variables that compose the original matrix. We used a principle component analysis because of the high association between these enzymes which compete for the same precursor molecule (*Page & Boubakir, 2014*; *Page & Stout, 2017*), have similar chemical structures (*Brenneisen, 2007*; *Flores-Sanchez & Verpoorte, 2008*) and genetic sequences (*Onofri, De Meijer & Mandolino, 2015*; *Vergara et al., 2019*), and may exemplify "sloppy enzymes" (*Auldridge, McCarty & Klee, 2006*; *Chakraborty et al., 2013*; *Franco, 2011*). Our results show that despite the evident competition for the same precursor, as seen with the negative correlation between THC and CBD (Fig. S2C), all of these compounds can be present together. Additionally, CBG is always seen in lower levels when compared to THC and CBD (Fig. S2), implying that THCA and CBDA synthases are efficiently converting CBGA into THCA and CBDA respectively in this population.

Studies suggest that THC has been selected for by breeders and growers and that varieties have been bread for higher THC potency (*ElSohly et al., 2016*; *Volkow et al., 2014*). Our results confirm these studies given that THC is always produced in higher quantities than CBD (Fig. S2), implying that THCA synthase may be a better competitor than CBDA synthase in this population.

Variation in THC production is probably a result of gene sequence variation (*Onofri, De Meijer & Mandolino, 2015*), expression levels, and gene copy number variation (*Vergara et al., 2019*), and there are multiple genes throughout the genome associated with its production (*Grassa et al., 2018*; *Laverty et al., 2019*). However, expression of these genes could be due to environmental effects such as cultivation conditions (*Elzinga et al., 2015*), which have not yet been quantified. Even though the parent plants were grown under different conditions than the BC1 offspring, all of the offspring were grown under the same conditions minimizing the environmental effects on the expression of these genes.

Although some correlations among traits are significant and make biological sense, the traits that are associated with purported groups (i.e., *indica* and *sativa*) within *Cannabis* are not correlated because of shared genetic basis. Therefore, trait correlations observed are due to either shared ancestry, in the case of comparisons among subspecies or other major lineages, or correlated selection, in the case of modern hybrids. In other words, correlations between leaf shape and phytochemistry may not be due to causal relationships, but rather because breeders have intentionally (or unintentionally) selected for these trait combinations. If these traits were associated due to shared ancestry or correlated selection, their association can be broken by recombination.

This is particularly noticeable in most of the modern cultivars which are hybrids from the supposed two main groups. Therefore, our study also suggests that common assumptions about associations between leaf shape and chemistry may exacerbate the misnaming problems of *Cannabis* varieties by the industry (*Sawler et al., 2015*; *Vergara et al., 2016*). Given the lack of association between cannabinoids and other morphological traits, the accepted standards for categorizing *Cannabis* types by the industry are deeply flawed because their naming convention is based on sets of traits that could be disassociated to each other. Additionally, other studies have shown that name is not indicative of cannabinoid potency or overall chemical composition (*Elzinga et al., 2015*), and that varieties are grouped based on reported flavors and aromas, regardless of genetic relationships misclassifying closely related individuals (*De la Fuente et al., 2020*). This misnaming problem in the *Cannabis* industry for both varieties and groupings ("sativa" and "indica") is greatly magnified by the fact that scientists can only study the *Cannabis* produced by the federal government despite its inferiority in potency and diversity, and the fact that it does not reflect the products distributed in consumer markets (*Schwabe et al., 2019*; *Vergara et al., 2017a*). It is crucial for *Cannabis* researchers to disseminate accurate information to the public. This is not being done adequately because the scientific literature is not effectively informing public policy, medical decisions, or providing correct information on harm reduction (*Hutchison et al., 2019*). This lack of information has major ramifications for growers, breeders, regulators, and consumers, particularly for medical patients who must understand what they are consuming to achieve the greatest benefit for their individual needs.

In order to improve the quality and efficacy of the *Cannabis* consumed by medical patients, it is important that unbiased, accurate, and precise chemotype testing should be made mandatory. However, testing facilities do not have universally established standards, as cannabinoid measurements vary widely across laboratories (*Jikomes & Zoorob, 2018*), and there are no supervising institutions that oversee testing entities or their methodologies, making differences in cannabinoid reporting inevitable.

## CONCLUSIONS

The fact that most of the phenotypic traits are not genetically correlated has significant implications for both *Cannabis* breeders and commercial growers. If these traits are not linked, as previously thought, then it is possible to select for new combination of traits when breeding for novel varieties. This expands the possibility of generating varieties with a unique combination of traits providing unforeseen medicinal and industrial value. Future breeding can be done to maximize combinations of these traits.

## ACKNOWLEDGEMENTS

We would like to thank J. A. Fuentes for help with the morphometric analysis, figure construction and for comments on the manuscript, to C. Pogoda, A. Schwabe, and R. Miller for comments on the manuscript, and to A. Holloway and two anonymous reviewers for their informative comments which greatly improved this paper.

### Funding

This research was supported by the Centennial Seeds brand, The Institute of Cannabis Research from Colorado State University Pueblo, by donations to the Agricultural Genomics Foundation and to the University of Colorado Foundation gift fund 13401977-Fin8 to Nolan C. Kane, and is part of the joint research agreement between the University of Colorado Boulder and Steep Hill Inc. The funders had no role in study design, data collection and analysis, decision to publish, or preparation of the manuscript.

### Grant Disclosures

The following grant information was disclosed by the authors:
Colorado State University Pueblo.
University of Colorado Foundation: 13401977-Fin8.

### Competing Interests

Daniela Vergara is the founder and president of the non-profit organization Agricultural Genomics Foundation, and the sole owner of CGRI, LLC; Ben Holmes is the owner of the Centennial Seeds brand; Jacob A. Haas is employed by DabLogic, LLC and a shareholder in LucidMood; Nolan C. Kane is a board member of the non-profit organization Agricultural Genomics Foundation.

### Author Contributions

- Daniela Vergara conceived and designed the experiments, performed the experiments, analyzed the data, prepared figures and/or tables, authored or reviewed drafts of the paper, bioinformatic code, and approved the final draft.
- Cellene Feathers performed the experiments, analyzed the data, prepared figures and/or tables, authored or reviewed drafts of the paper, bioinformatic code, and approved the final draft.
- Ezra L. Huscher performed the experiments, analyzed the data, prepared figures and/or tables, authored or reviewed drafts of the paper, bioinformatic code, and approved the final draft.
- Ben Holmes conceived and designed the experiments, performed the experiments, authored or reviewed drafts of the paper, and approved the final draft.
- Jacob A. Haas performed the experiments, authored or reviewed drafts of the paper, and approved the final draft.
- Nolan C. Kane conceived and designed the experiments, performed the experiments, authored or reviewed drafts of the paper, and approved the final draft.

### Data Availability

Raw data and code are available in the Supplemental Files. They are also available at Dryad (https://doi.org/10.5061/dryad.6t1g1jwxh) and GitHub (https://github.com/KaneLab/daniela_cannabis/blob/master/Vergara_et_al_phenotypic_code_07082020.txt).

## Supplemental Information

Supplemental information for this article can be found online at http://dx.doi.org/10.7717/peerj.10672#supplemental-information.

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
