# Peer review of "Widely assumed phenotypic associations in Cannabis sativa lack a shared genetic basis"

_PeerJ, doi:10.7717/peerj.10672_

## Round 0.1 · original submission · Minor Revisions

Please address critiques of all reviewers and amend your manuscript accordingly.

·

Basic reporting

no comment - nicely done - just have recommendations in next section for improving clarity

Experimental design

Recommend setting up the thesis of the paper more clearly. I see two main points: 1) Traits that are correlated may share common developmental pathways which would inhibit selecting for new combinations of traits. The authors investigated this by assessing the correlation in traits at early and later developmental time points and found that traits were not correlated, for the most part, and are likely inherited independently. The authors hit on this in several places, including the Conclusions section, but I would like to see the hypothesis clearly set up in the introduction.

2) Many commercially important traits in cannabis are expressed at maturity. If breeders could predict late stage expression via correlations with traits expressed earlier during development, selections could be made sooner and breeding cycles accelerated. The authors did the analysis to test this hypothesis (discussed on pg 13-14), but I would like to see the hypothesis set up in the introduction and then discuss whether the hypothesis is supported or rejected. Similarly, the authors bring up the important point about identifying male and female plants - if early traits distinguished the sexes, males could be culled before contaminating field with pollen and reducing yield. Unfortunately, this doesn't appear to be the case, but their findings are valuable and bear reporting!

There is also the question of classification of plants. I don't see this as a major point of the paper since it has been covered extensively in other literature. However, it is relevant to introduce and re-emphasize that historical classifications and associated nomenclature are not supported by this investigation or other scientific work. Authors discuss findings related to this on p14 line 282-285, but need to give context as to why the findings are important.

Validity of the findings

p14, line 242-245 - seems like there are other explanations for lack of correlation between timepoints - were there differences in flower initiation time or maturation time? some plants "stretch" more than others - i.e., they grow more during the flowering stage than others - and there are also differences in apical dominance between varieties, which impacts the relationship between height and #nodes.

Additional comments

Minor edits

1. p6, line 47. need to add "possibly" or "may be" before "causally associated" since you aren't testing that specifically.
2. p6, line 62-63. wild populations that share ancestral traits would have a common ancestor and would not inherit independently.
3. p7, line 79-80. cite Laverty et al. and the BioRxiv papers that show the physical location of cannabinoid synthase genes
4. p8, line 98-100. cite Clarke and Merlin for the lineage definitions.
5. p13, line 217 add links
6. p20, line 371. THCA synthase may be a better competitor than CBDA synthase for the CBGA precursor, but THCAS could also be expressed at higher levels than CBDAS in these plants. Are there expression data that could help disambiguate?
7. p21, line 390. clarify what "this" is referring to in the introductory sentence
8. p21, replace "strain" with "variety" or "cultivar" as appropriate
9. p21, line 406-408. I agree that we need to do a better job of getting information out to the public and to make sure that we're doing all we can to make sure that research informs policy, medicine, etc.! Maybe break these two points into two sentences for clarity.

Reviewer 2 ·

Basic reporting

The manuscript is well written and describes all the methods clearly. The authors have looked extensively at the phenotypic and morphological changes in the inter-crossed lineages and performed correlation analysis.
Some concerns:

1) The authors need to change the title as they are not looking at any gene analysis of the species.
2) Grammatical errors in line 154 and line 368. Please check.

Experimental design

The experiments are well set up extensive endpoints have been measured.
It will be helpful if you can justify why you backcrossed male with female F1 offspring of the cross.

Validity of the findings

The authors have found some interesting results and point out a gap in the way cannabis nomenclature is adopted. The results can be further very useful in studies related to crossbreeding techniques in cannabis.

Reviewer 3 ·

Basic reporting

Yes, meets the criterion.

Experimental design

Well designed research.

Validity of the findings

Data supported by valid evidence based research.

Additional comments

The manuscript entitled "Widely assumed phenotypic associations in Cannabis sativa---." is an important study conducted by the authors. The manuscript is well written, and research is well supported by data.
The manuscript can be accepted after minor revision.
Few typo errors need to be fixed.
Line 95 Give a period after the sentence 2012).
Line 218 R studio, give space
Line 468 D uppercase for difference
Line 500 G uppercase for geomorph

---

## Round 0.2 · accepted · Accept

Since all issues were addressed and the manuscript was revised accordingly, the amended version is acceptable now.